# A Novel Process-Oriented Graph Storage for Dynamic Geographic Phenomena

**Cunjin Xue [1,*], Chengbin Wu [1], Jingyi Liu [1] and Fenzhen Su [2,*]**

[1]   Key Laboratory of Digital Earth Science, Aerospace Information Research Institute, Chinese Academy of Sciences, Beijing 100094, China; wcb892534877@icloud.com (C.W.); liujy2017@radi.ac.cn (J.L.)

[2]   Institute of Geographical Science and Natural Resources Research, Chinese Academy of Sciences, Beijing 100101, China

*   Correspondence: xuecj@radi.ac.cn (C.X.); sufz@lreis.ac.cn (F.S.); Tel.: +86-10-82178126 (C.X.); Tel.: +86-10-64888956 (F.S.)

**Abstract:** There exists a sort of dynamic geographic phenomenon in the real world that has a property which is maintained from production through development to death. Using traditional storage units, e.g., point, line, and polygon, researchers face great challenges in exploring the spatial evolution of dynamic phenomena during their lifespan. Thus, this paper proposes a process-oriented two-tier graph model named *PoTGM* to store the dynamic geographic phenomena. The core ideas of *PoTGM* are as follows. 1) A dynamic geographic phenomenon is abstracted into a process with a property that is maintained from production through development to death. A process consists of evolution sequences which include instantaneous states. 2) *PoTGM* integrates a process graph and a sequence graph using a node–edge structure, in which there are four types of nodes, i.e., a process node, a sequence node, a state node, and a linked node, as well as two types of edges, i.e., an including edge and an evolution edge. 3) A node stores an object, i.e., a process object, a sequence object, or a state object, and an edge stores a relationship, i.e., an including or evolution relationship between two objects. Experiments on simulated datasets are used to demonstrate an at least one order of magnitude advantage of *PoTGM* in relation to relationship querying and to compare it with the Oracle spatial database. The applications on the sea surface temperature remote sensing products in the Pacific Ocean show that *PoTGM* can effectively explore marine objects as well as spatial evolution, and these behaviors may provide new references for global change research.

**Keywords:** geographic process; dynamic phenomena; graph-based storage model; spatial evolution; sea surface temperature anomalies

## 1. Introduction

The geographic world is dynamic [1]. There is a large number of dynamic geographic phenomena in the physical world, e.g., the variation of an ocean eddy or front, the evolution of a typhoon or a rainstorm, the dynamic behavior of an urban heat island, the spread of a wildfire, the propagation of a disease, etc. Advanced remote sensing and crowdsourcing survey technologies provide a foundation for obtaining these dynamic phenomena and make it possible to analyze and discover their spatiotemporal patterns and evolutions [2,3]. To effectively analyze and discover the spatiotemporal patterns of these dynamic phenomena, an appropriate spatiotemporal data model is needed to represent and organize them [4–6].

A spatiotemporal data model has gone through three phases, i.e., a static model, an object-based or an event-based changing model, and a process-oriented simulating model. The static spatiotemporal data model, e.g., a snapshot-based data model, a space-time cube model, or a space-time composite

model, takes the temporal dimension as an attribute of a location represented by two/three geographic coordinates [7]. Thus, continuous phenomena are traditionally represented as an ordered set of independent snapshots or as a three-dimensional spatiotemporal structure [8]. The object/event-based spatiotemporal data model views the entity as an occurrence not a continuant [9–12]. That is to say, this model is discrete and not capable of dealing with the continuous phenomena [13]. For a dynamic geographic phenomenon, the spatiotemporal data model depicts not only the entities themselves but also the spatial evolution during their lifespans [14]. As it has a clear lack of appropriate structures for handling the spatial evolution, the object/event-based data model has a great challenge to deal with, i.e., the contradiction between the discrete entities and continuous phenomena [15]. For a better understanding of the behavior of dynamic phenomena and the underlying mechanisms driving movement, the process-oriented spatiotemporal data model for handing the dynamic phenomena has obtained more attention in recent years [8,16–18].

The process-oriented data model concerns the consistency of the space, time, and attributes among the evolution entities in the conceptual design [19–21]. Yuan designed a hierarchical framework where moving objects were described by four different states and a zone-sequence-process-event [22]. A zone is defined as an area of spatial contiguous cells meeting a designated threshold in a snapshot; a sequence is a set of zones meeting a threshold in consecutive snapshots; a process is a set of sequences where the geographic objects are topologically related; and an event includes at least one type of process during a consecutive time interval. Xue et al. proposed a process-oriented dual representing framework with a semantics of hierarchical abstraction and an object of being included by level, i.e., a process-sequence-phase-state [16]. The state focuses on the spatial information; the sequence and phase do the temporal information; and the process integrates the spatial, temporal, and thematic information. Yi et al. used a hierarchy of static structure-process-scenario to track, query, and evaluate the ocean eddies in the South China Sea [23]. In these models, the included relationships are clear in the time scale, e.g., an event includes at least one process, the process includes two or more sequences, and a sequence includes two or more states, while the evolution relationships between the consecutive snapshots are missing [22]. That is, these models concern the objects and their changes not the behaviors causing the changes, which limits further analysis, especially that of the splitting or merging behaviors between the consecutive snapshots.

For addressing the evolution, change, and interaction of graphical entities, Mondo et al. proposed a graph-based spatiotemporal model [14,24]. The nodes store spatially contiguous entities at each time step, and the direct edges store the spatial and temporal linkages among graphical entities at two adjacent snapshots, where the direction indicates the time sequence. Considering the entities, their changes, and the cause of their simultaneous changes, the graph-based model has been widely used to explore dynamic geographic patterns in recent years. For example, Liu et al. used the event-based model to represent rainfall storms and then designed directed spatiotemporal graphs to characterize their initiation, development, movement, and cessation [25]. Zhu et al. modeled the urban heat island as a spatiotemporal field-object with its own life-cycle and designed a graph structure with a zone and sequence to reconstruct its evolutionary process [26]. The event-based and field-object representations make it difficult to describe the uniform relationships between entities.

A common way to store such types of dynamic graphical entities is an object-relational database. However, these spatial databases, e.g., Oracle spatial, SQLSever, and Postgres, focus on the storage and management of objects not their spatial changes. To improve their abilities in dealing with the temporal relationships between consecutive snapshots, the spatial object database will add logistical relationship tables to the database [16]. The more complex the temporal relationship is, the more complex the logistical relationship table is, and the fewer the abilities of dealing with the temporal relationship are. Compared to the object-relational database, the graph database uses the graph-based model to store the geographic objects and their evolution relationships, i.e., the node stores the objects, and the edge stores the relationships [24]. Using an index-free adjacency to describe the relationships

between entities [27], the graph-based model performs better in dealing with the spatial evolution of dynamic graphical entities than the object-relational database does.

Thus, the purpose of our study is to design a spatiotemporal graph-based model for dealing with the dynamic geographic phenomena. The paper is organized as follows. Section 2 discusses the dynamic geographic phenomena and their process-oriented semantics. Section 3 summarizes the four types of nodes and two types of edge and designs a two-tier process graph model, i.e., a process-oriented graph and sequence-oriented graph. Two case studies, with a simulated dataset and a real dataset, are addressed in Section 4, while Section 5 presents our discussions and conclusions.

## 2. Process-Oriented Geographic Semantics

### 2.1. Dynamic Geographic Phenomena

A phenomenon with a continuously changing property from production through development to death is defined as a process in this paper. Thus, the process also has an evolutionary property that is obviously different to discrete changes in relation to the following issues:

- Thematic changes with time: a continuous change vs. an instantaneous change, shown in Figure 1, from T1 to T2, from T2 to T3, or from T3 to T4;
- Thematic changes with spatial locations: changes vs. no changes, shown in Figure 1, from each snapshot;
- Spatial changes with time: a continuous change vs. an instantaneous or no change [20], shown in Figure 1, from T1 to T2, from T2 to T3, or from T3 to T4; and
- Evolution: a process has a lifespan, while a discrete change occurrence means termination.

Figure 1 shows the differences in detail. The different colors represent the intensities of thematic information, and the geometric shape and location mean the changes of spatial information. At each time snapshot from T1 to T2, from T2 to T3, or from T3 to T4, e.g., T11, T12, T21, T22, T31, and T32, there is no change in Figure 1a, while Figure 1b shows a continuous change in thematic and spatial information.

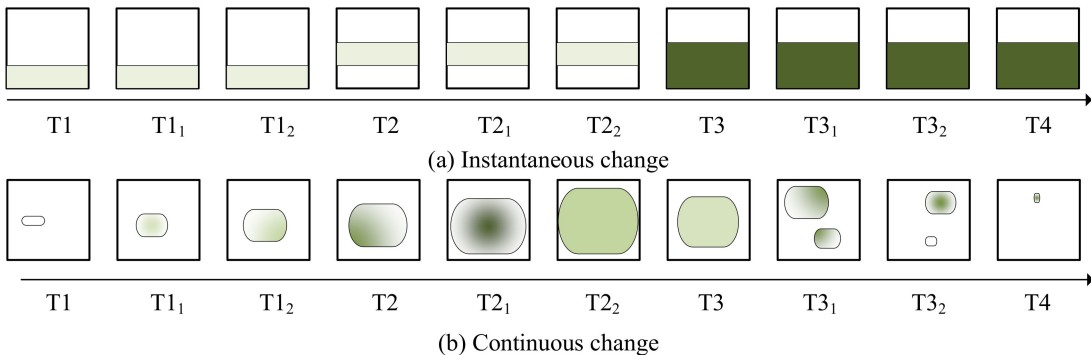

**Figure 1.** Comparisons of the (**a**) instantaneous and (**b**) continuous changes.

### 2.2. Geographic Process Semantics

This paper uses a hierarchical framework of "geographic process-evolution sequence-instantaneous state" to represent a dynamic phenomenon. Figure 2 shows the concept model of the geographic process.

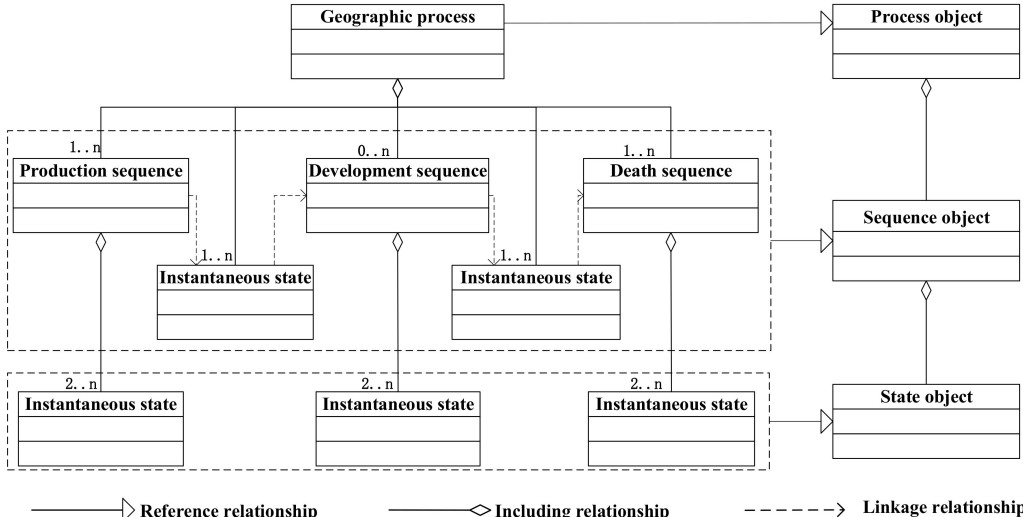

**Figure 2.** The concept model of the geographic process.

The geographic process depicts an evolution from production through development to death as a whole, which consists of one to several evolution sequences. The changes within the process not only include thematic information but also the spatial structure of the dynamic phenomenon.

The evolution sequence consists of two or more instantaneous states during the lifespan, and each sequence has a similar spatial structure and similar thematic characteristics. According to an evolution property, there are three types of basic sequence, i.e., a production sequence, a development sequence, and a death sequence. The basic sequences are linked together through instantaneous states to construct a complicated sequence or a geographic process.

- Production sequence: A sequence begins to strengthen in the spatial structure and thematic characteristics from the generation of the geographic process.
- Development sequence: A sequence has a similar spatial structure and similar thematic characteristics during a lifespan of the geographic process, where its start/end time is not a generation/termination snapshot of the geographic process. The similar behavior means continuing to strengthen, continuing to weaken, or continuing to remain stable.
- Death sequence: A sequence weakens in the spatial structure and thematic characteristics, where its end time is a termination snapshot of the geographic process.

An instantaneous state depicts a spatial structure and the thematic characteristics of a geographic phenomenon at a given snapshot. According to a relationship between consecutive snapshots within the process, we summarize six types of instantaneous states. Denoted as $S_{t-1}$, $S_t$, and $S_{t+1}$ as the instantaneous states at the three consecutive snapshots, t-1, t, and t+1, respectively, this paper defines the six instantaneous states as follows, and Figure 3 gives their diagrams.

- Production state: if $S_{t-1}$ does not exist and $S_t$ exists, then $S_t$ is a production state;
- Development state: if $S_{t-1}$ and $S_{t+1}$ are the exclusive ones that exist and $S_t$ exists, then $S_t$ is a development state;
- Merging state: if two or more $S_{t-1}$ exist, $S_t$ exists, and $S_{t+1}$ is the exclusive one that exists, then $S_t$ is a merging state;
- Splitting state: if $S_{t-1}$ is the exclusive one that exists, $S_t$ exists, and two or more $S_{t+1}$ exist, then $S_t$ is a splitting state;
- Merging-splitting state: if two or more $S_{t-1}$ exist, $S_t$ exists, and two or more $S_{t+1}$ exist, then $S_t$ is a merging-splitting state;
- Termination state: if $S_t$ exists and $S_{t+1}$ does not exist, then $S_t$ is a termination state.

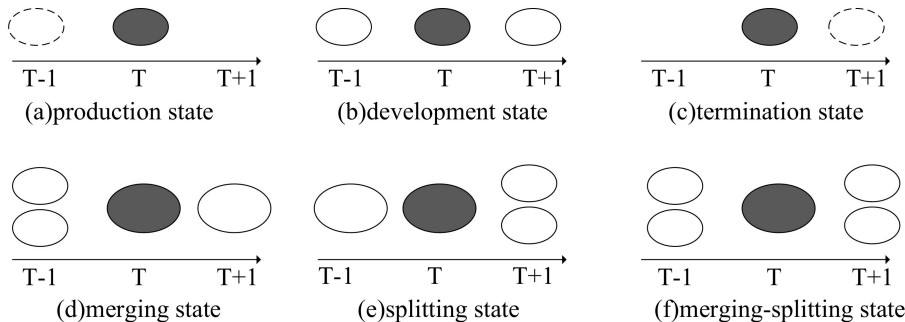

**Figure 3.** The six types of instantaneous states.

## 2.3. Relationships of the Geographic Process

There are three kinds of relationships within a geographic process, i.e., a spatiotemporal relationship, an including relationship, and an evolution relationship. A spatiotemporal relationship depicts a spatiotemporal distance, a topology, and a direction between geographic processes, between evolution sequences, or between instantaneous states. An including relationship depicts a part–whole relationship of a geographic process, meaning that a geographic process includes one or more evolution sequences and that an evolution sequence includes two or more instantaneous states. An evolution relationship depicts the motivation within the process, indicating how to transform from the previous snapshots to the current one and from the current to the next. According to the instantaneous state types in the previous and next snapshots, this paper summarizes four types of evolution relationships, i.e., merging, splitting, development, and splitting–merging relationships. Figure 4 gives their diagrams.

- A merging relationship represents an interaction between two or more objects in previous snapshots that merge one object with the current one. The previous snapshot is a production, development, or merging state, and the next is a merging or merging-splitting state. The relationship between them is a merging relationship.
- A splitting relationship represents an interaction between one object in the current snapshot splitting two or more objects in the next one. The previous state is a splitting or merging-splitting state, and the next is a production, splitting, or termination state. The relationship between them is a splitting relationship.
- A development relationship represents no interaction with other objects and one object that moves from the previous to the current and then to the next snapshot. The previous state is a production, development, or merging state, and the next is a production, splitting, or termination state. The relationship between them is a development relationship.
- A splitting–merging relationship represents an interaction between a part of one object and a part or whole of another object in the previous snapshot, merging into a new object in the current one. The previous snapshot is a splitting or merging-splitting state, and the next is a merging or merging-splitting state. The relationship between them is a splitting–merging relationship.

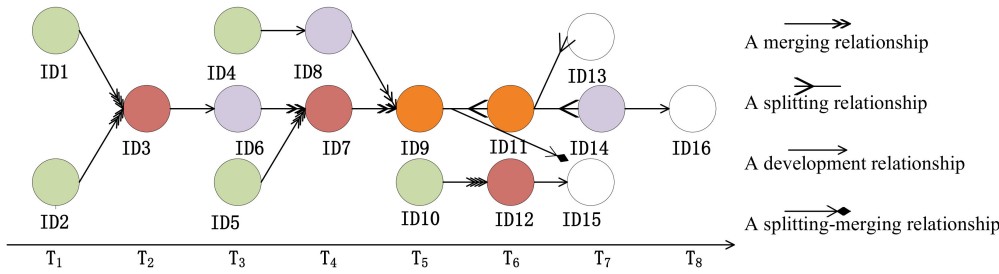

**Figure 4.** The four types of evolution relationships between instantaneous states.

### 3. Process-Oriented Two-Tier Graph Model

$G(N, E)$ is the formal expression of a graph, where $N$ represents a node, storing a geographic entity, and $E$ represents an edge, storing a relationship between two geographic entities. Here, an entity may be a process, a sequence, or a state. As mentioned above, a process consists of one or more basic evolution sequences, and the basic sequences are linked together through instantaneous states to construct the geographic process. These instantaneous states, responding to linking the basic sequences, are defined as linked nodes. Thus, in a process-oriented graph model, there are four types of node, i.e., a process node, a sequence node, a state node, and a linked node. According to process semantics, there are three kinds of relationships among processes, i.e., a spatiotemporal relationship, an including one, and an evolution one. As a spatiotemporal relationship could be calculated in real time through geographic entities, a node stores it implicitly, e.g., a spatiotemporal distance, direction, and topology. Thus, there are two types of edge in our proposed model; one edge stores an including relationship, and the other stores an evolution relationship.

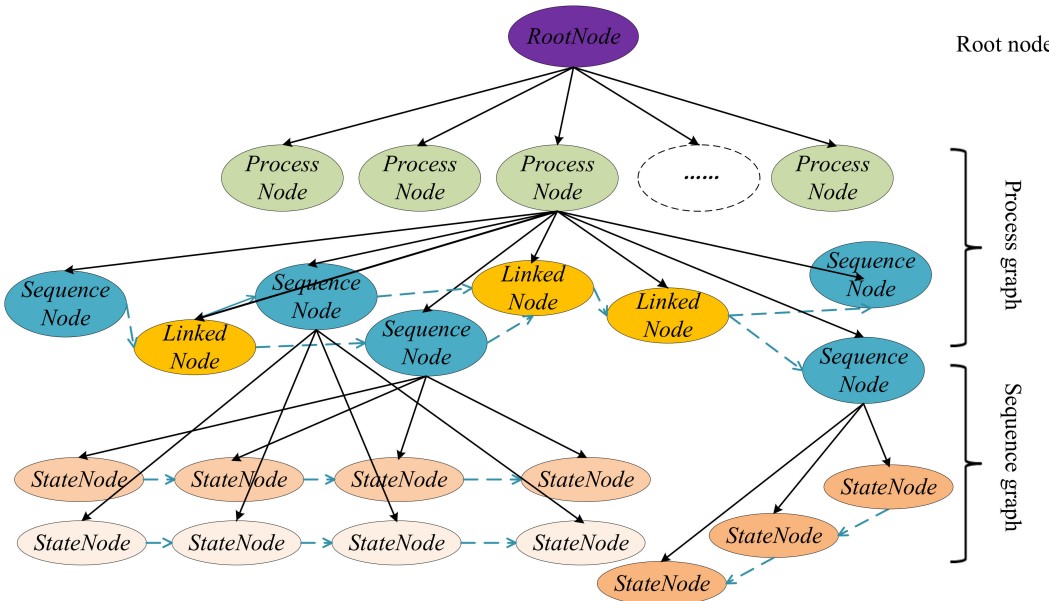

**Figure 5.** A process-oriented two-tier graph model: The solid line represents the including relationships, and the dot line represents the evolution relationships.

To store four types of node and two types of edge, this paper designs a process-oriented two-tier graph model named *PoTGM* which includes a process graph and a sequence graph, as shown in Figure 5. The process graph is the first-layer graph in which the nodes consist of process nodes, sequence nodes, and linked nodes, and the relationships consist of the including ones, between the process nodes and sequence nodes and the evolution ones and between the sequence nodes and the linked nodes. The sequence graph is a second-layer graph which consists of sequence nodes and state nodes and consists of the including relationships between the sequence nodes and state nodes and the evolution relationships between the state nodes. In the *PoTGM*, each sequence node in a process graph is represented and stored by a sequence graph.

### 3.1. A Process Graph

A process graph stores three types of node, i.e., a process node, a sequence node, and a linked node, denoted as *ProcessNode*, *SequenceNode*, and *LinkedNode*, respectively. *ProcessNode* stores the whole dynamic geographic phenomenon, including its spatiotemporal structure and thematic information, and is the father of *SequenceNode* and *LinkedNode*. *SequenceNode* stores the spatiotemporal structure and thematic information of an evolution sequence. *LinkedNode* is responsible for linking the different

evolution sequences. In the process graph, *LinkedNode*, generally, is a merging, a splitting or a merging-splitting state, which has a dramatic change in the spatial or temporal domain. An edge between the *ProcessNode* and *SequenceNode* or *LinkedNode*, denoted as the *IncludingEdge*, stores an including relationship, and an edge between the *SequenceNode* and *LinkedNode*, denoted as the *EvolutionEdge*, stores an evolution relationship. Using a key–value pair, Figure 6 gives the storage of a process graph.

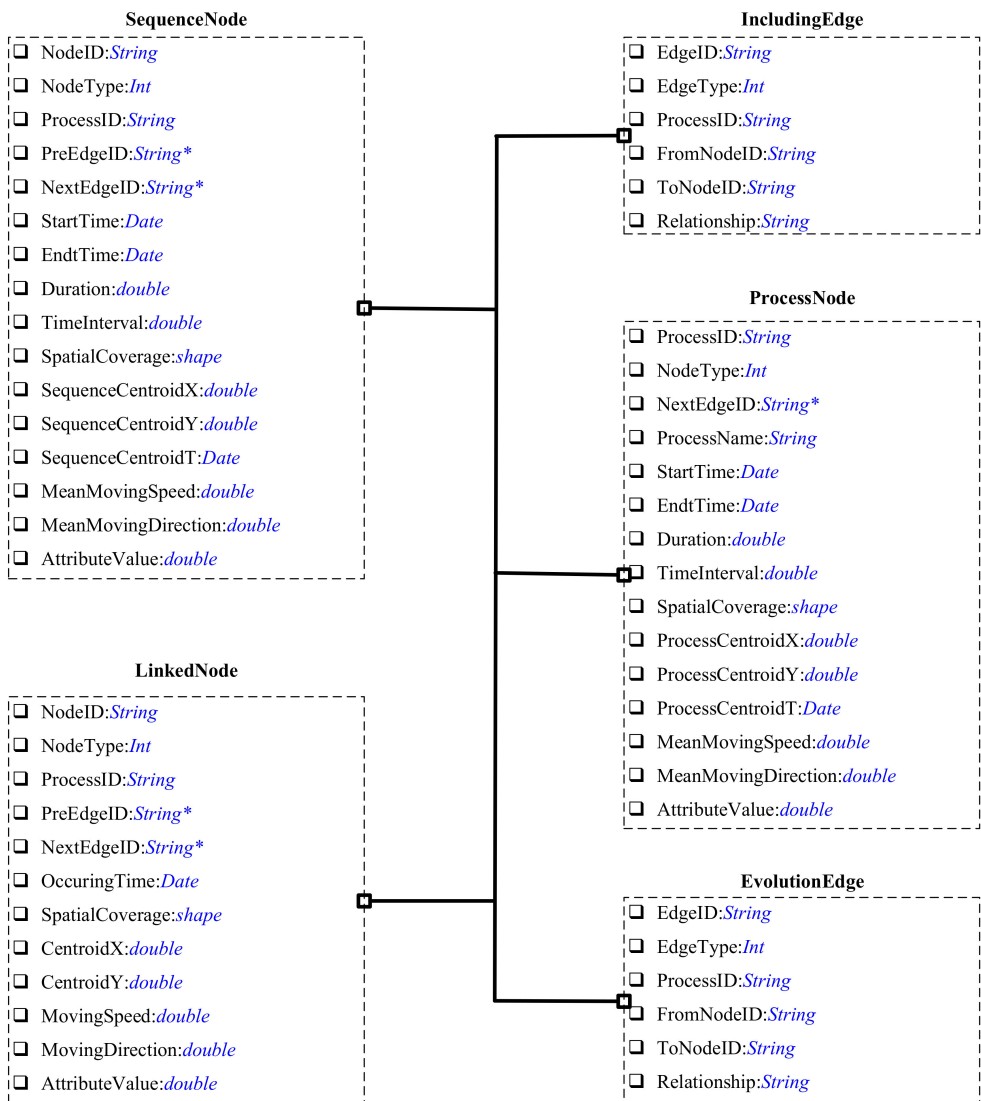

**Figure 6.** The storage of a process graph A: The process is linked by *NodeID* and *EdgeID*.

In a process graph, through the *ProcessNode*, *SequneceNode*, and *LinkedNode*, we could obtain the spatiotemporal structure and thematic characteristics of a process, an evolution sequence, and an instantaneous state, respectively. Through the *IncludingEdge*, a process could find its evolution sequences and instantaneous states, and vice versa, and an evolution sequence or an instantaneous state could also find a process that it belongs to. Through the *EvolutionEdge*, a process could find all the dynamic behaviors among its sequences and states, and an evolution sequence or an instantaneous state could also obtain its previous and next sequence or state.

### 3.2. A Sequence Graph

A sequence graph consists of two types of node, a sequence node and a state node, denoted as the *SequenceNode* and *StateNode*, and two types of edge; one is an including relationship, and the other is an evolution one, denoted as the *IncludingEdge* and *EvolutionEdge*, respectively. The *SequenceNode* here stores the same information as does a process graph, which is the father of the *StateNode*. The *StateNode* stores spatial and thematic information on an instantaneous state. The *IncludingEdge* stores the relationship between the *SequenceNode* and *StateNode,* and the *EvolutionEdge* stores the relationship between the *StateNodes.* Figure 7 gives the storage of a sequence graph.

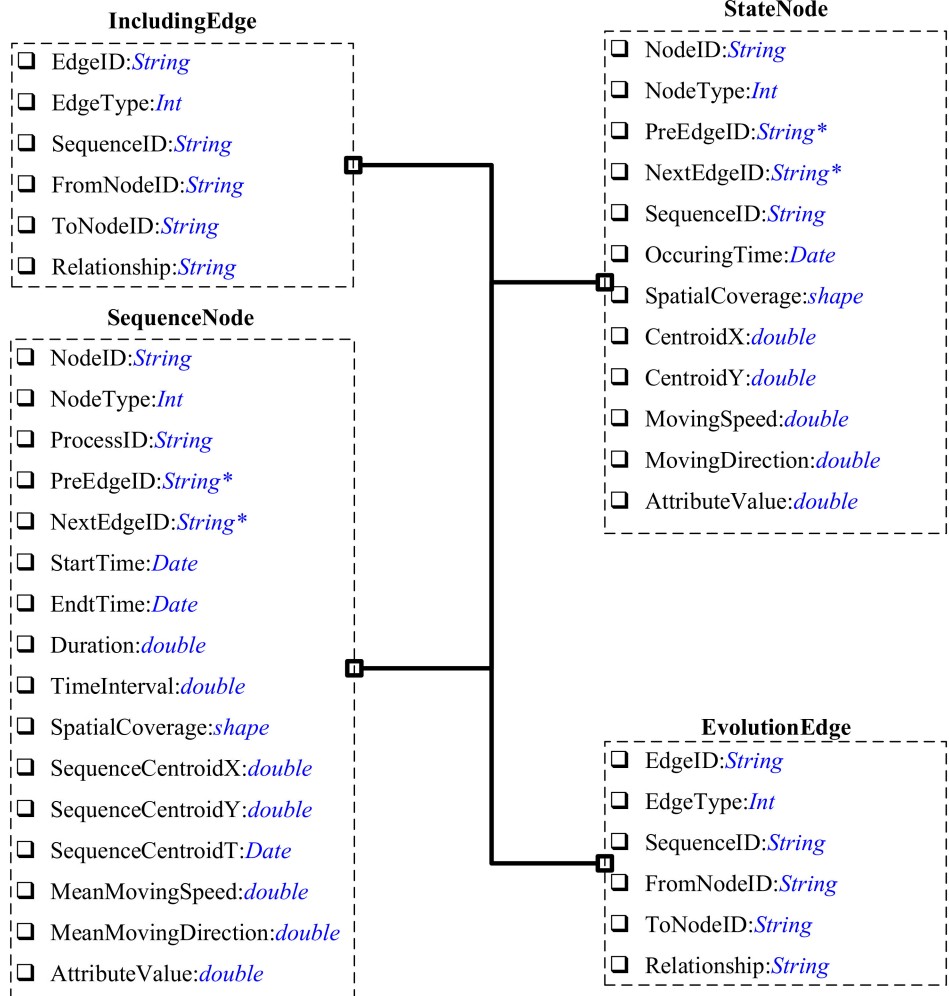

**Figure 7.** The storage of a sequence graph A: The sequence is linked by *NodeID* and *EdgeID*.

The *SequenceNode* and *StateNode* obtain the spatiotemporal structure and thematic characteristics of an evolution sequence and an instantaneous state. The *IncludingEdge* finds an including relationship between a sequence and all of its instantaneous states, and the *EvolutionEdge* finds a dynamic behavior in a snapshot state, transformed from the previous to the current and from the current to the next.

For example: To illustrate the process-oriented two-tier graph model, this paper simulates a geographic process as an example. The simulated process includes seven evolution sequences and twenty-five instantaneous states, as shown in Figure 8. Three of the twenty-five instantaneous states are modeled as *LinkedNodes*, and the others are *StateNodes*. Thus, there is one process graph and seven sequence graphs. The process graph stores one process object including seven sequence objects and three state objects. The sequence graph stores one sequence object including several state objects,

e.g., the sequence graph-based *Sequence 1* includes three state objects, and the sequence graph-based *Sequence 5* includes five state objects.

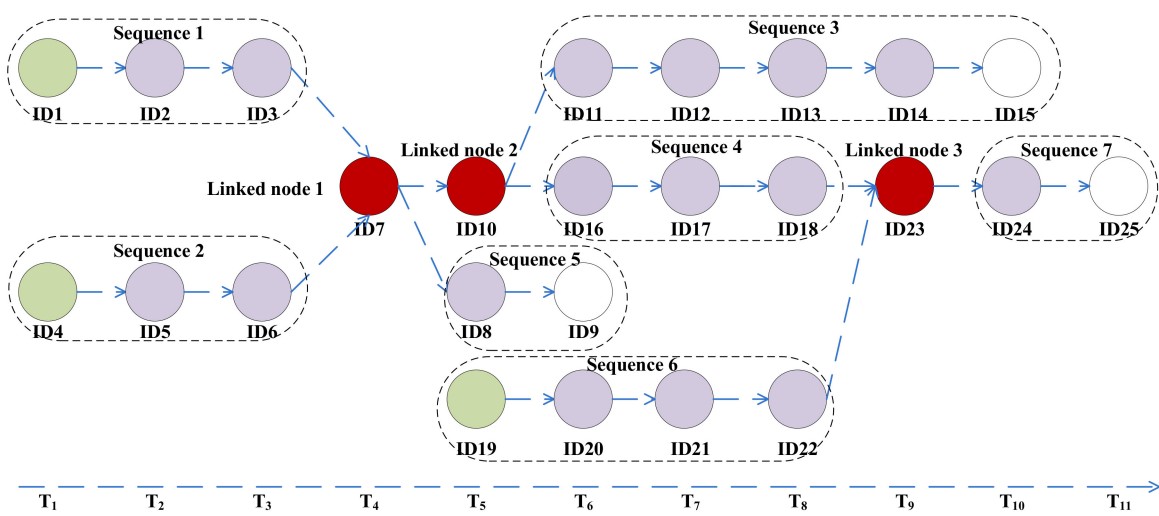

**Figure 8.** A simulated process.

### 3.3. Process-Oriented Graph Database Based on Neo4j

Neo4j is a graph database which consists of a node store file and a relationship store file. The former stores node records, and the latter stores relationship records. A node record and a relationship record are linked by doubly linked lists [27]. Being capable of index-free adjacency, Neo4j performs better on dynamic data storage and relationship queries than traditional relational databases, such as Oracle spatial, SQLSever, and Postgres. The index-free adjacency means that the adjacent nodes can be found by following the explicit pointers that connect the nodes without having to perform an index lookup. Figure 9 shows a schema based on the Neo4j of the *PoTGM* storage using the doubly linked list.

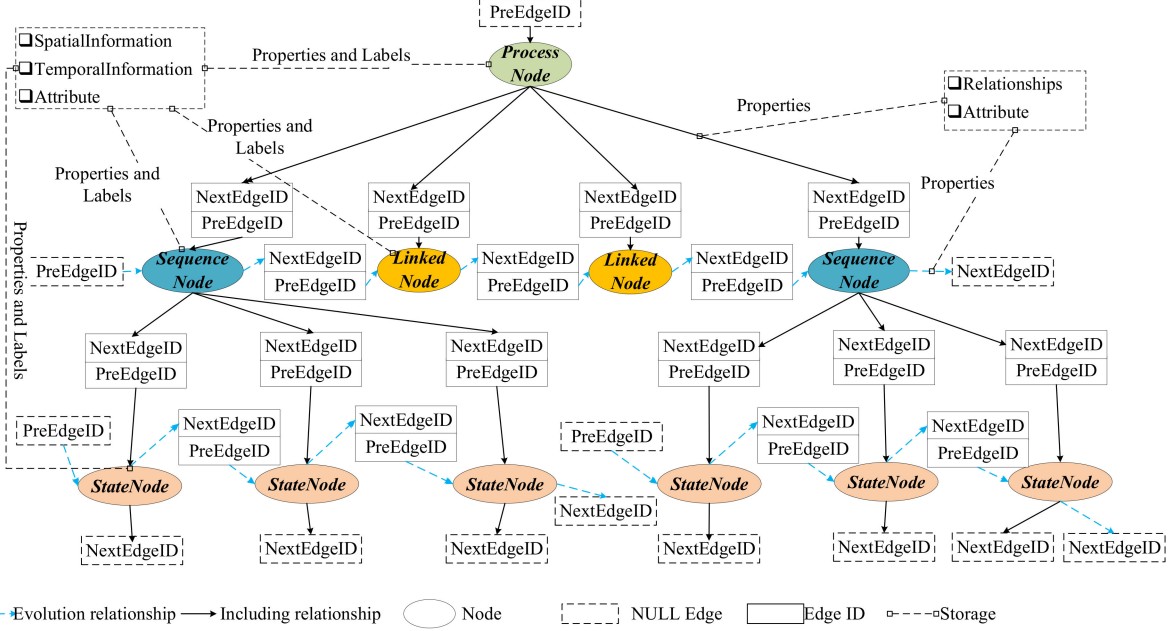

**Figure 9.** A schema of PoTGM based on Neo4j.

Using the doubly linked mechanism, each node stores one or more outputting edges and one or more inputting edges. Each edge links only two nodes. According to an edge direction, a previous node and a next node are defined. An edge stores two pointers; one points to the previous node, denoted as the *NextEdgeID,* and the other points to the next node, denoted as the *PreEdgeID*. If an edge only has a previous node or a next node, the edge does not exist. Correspondingly, the pointer outputting from a previous node points to NULL, or the pointer inputting to a next node points to NULL.

## 4. Evaluations and Discussions

### 4.1. Performance Analysis

This experiment uses 100; 1000; 10,000; 100,000; and 1,000,000 copies of the simulated data (shown as Figure 8) to build the databases. Oracle spatial, an object-relational database, is used to compare the results with Neo4j in relation to the database storage, object query, and relationship query. We follow Figure 9 to build the Neo4j database with version 4.1.3, and we follow Figure 10 to build the Oracle spatial database with version 11g. In the Oracle spatial database, there are three object tables, i.e., the *ProcessObjectTable*, *SequenceObjectTable*, and *StateObjectTable*, and two relationship tables, i.e., the *EvolutionRelationshipTable* and *IncludingRelationshipTable*, where they are linked together by object IDs, i.e., the *ProcessID*, *SequenceID*, and *StateID*. The experimental hardware environment includes a Xeon E5 core 12 CPU at 2.20 GHz, a 2400 GB hard disk, and 32.0 GB of memory.

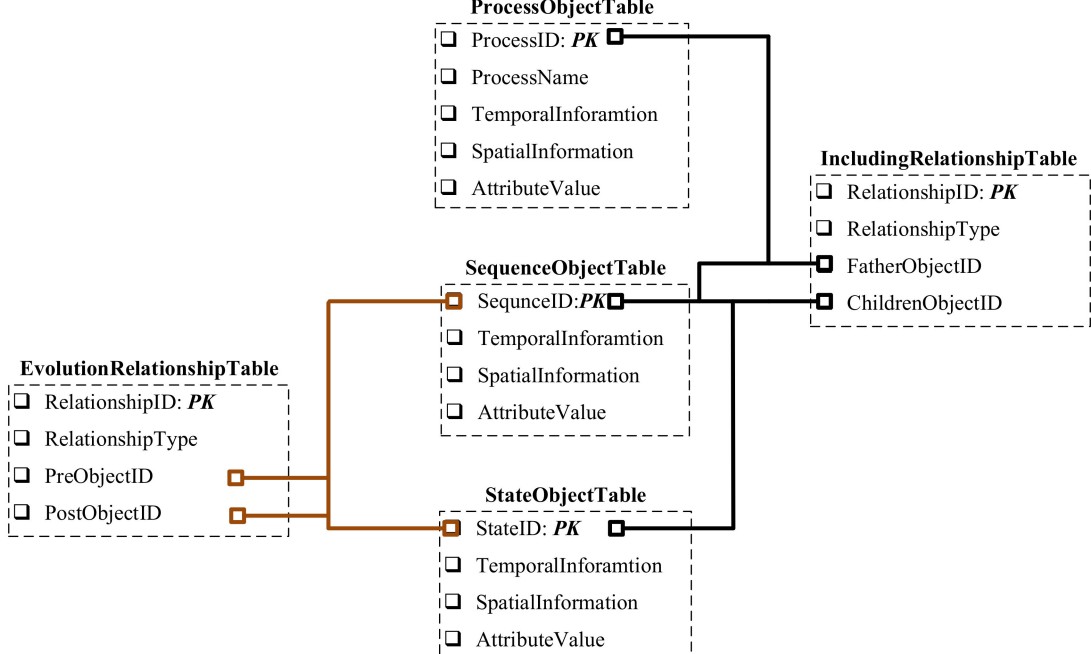

**Figure 10.** A schema of the object relational database: The black line links the process objects, and the orange line links the evolution relationships.

In relation to database storage capacity, Oracle spatial performs better than Neo4j. Figure 11a shows that their storage capacities all increase with the database size, i.e., a number of processes, and the Neo4j storage capacity is one order of magnitude higher than Oracle spatial. A proportion of Neo4j's storage capacities accounts for above 80%, and with an increase of database size, it will rise to 90% (Figure 11b). The reasons for this are that the Oracle spatial database stores regular tables, i.e., an object table and a relationship table, while the Neo4j database stores irregular files, i.e., a node store file and a relationship store file, and that Neo4j uses a doubly linked mechanism to

store the edges/relationships, while the Oracle spatial uses a table to store relationships. That is, the relationships are stored twice in Neo4j, while in Oracle spatial, they are stored only once.

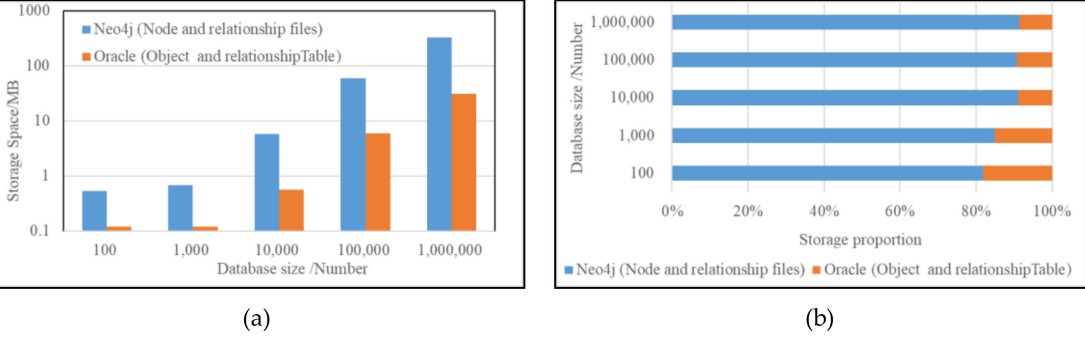

(a)                    (b)

**Figure 11.** Comparisons of the database storage capacities.

Regarding the performance of a spatial object query, Neo4j has a slight advantage due to its small database size. With an increase of the database size, Oracle spatial performs better than Neo4j. The reasons are that, after finding a matched object, the Neo4j still goes through the database one time to determine whether to update the object before returning it. The determination of whether to update or not is an unnecessary step for querying spatial objects, and this is, therefore, a debug of Neo4j. The Oracle spatial database will use a spatial index to find the matched object and to return it. Figure 12 gives their relative computation times involved in querying a given process object, including its sequence objects and state objects in different database sizes.

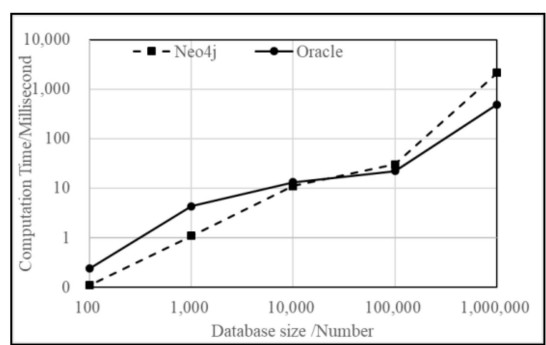

**Figure 12.** Comparisons of querying objects in different database sizes.

There are two types of evolution relationships among dynamic geographic phenomena. One is set under a given database size to query all relationships between consecutive snapshots. Here, we adopt the depth of the relationship [27] to depict the spatial evolution of an object through successive snapshots. Between two snapshots, the depth of the relationship is one; between three snapshots, the depth is two, i.e., a relationship-of-relationship; and between four, it is three, i.e., a relationship-of-relationship-of-relationship, etc. The other is set against different database sizes to find all the evolution relationships from one instantaneous state to another during its lifespan. Figure 13a illustrates the computation time of querying all of the relationships of the object from $T_4$ to $T_{10}$, as shown as Figure 8. The relationship depths are from one to six, where the database size is 1,000,000 process objects. Figure 13b illustrates the computation time of querying the relationships from object ID7 to ID23 in different database sizes. The results show that Neo4j performs much better than the Oracle spatial database in querying evolution relationships, and the performance advantage is enhanced with an increase of the depth of the relationship or database size. To query an evolution relationship with a depth of $n$, the Neo4j uses a *Next* function of nodes to go through the doubly linked lists $n$ times, where the computation complexity is $O(n)$, while the Oracle spatial database conducts the nested loop $n$ times to join the object-relational tables, where the computation complexity is $O(n*n)$.

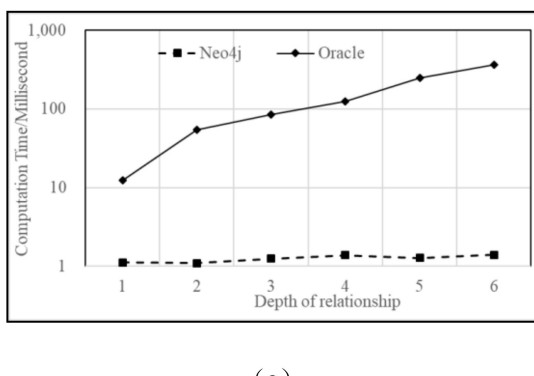 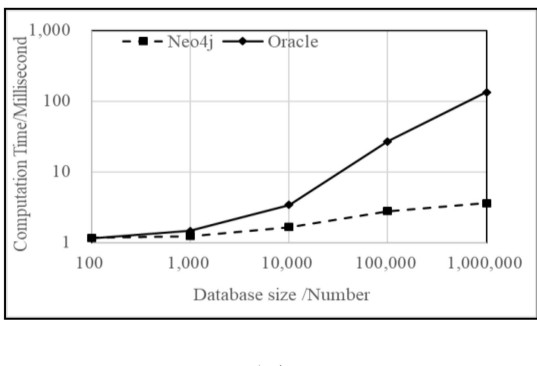

（a） （b）

**Figure 13.** Comparisons of the querying evolution relationships.

*4.2. Evolution of Abnormal Marine Variation and its Relationship with ENSO*

El Niño-Southern Oscillation (ENSO) is a typical signal of global climate variability that affects a variety of marine environmental parameters [28]. The Pacific Ocean, from 100 °E to 60 °W and 50 °S to 50 °N, where it is sensitive to global climate change and regional sea–air interactions and is responsible for marine variations, is considered as a research area. The test data are the sea surface temperature products for the period from January 1985 to December 2017, with a spatial resolution of 1° and a temporal resolution of 1 month. These data are obtained from the NOAA Optimum Interpolation Sea Surface Temperature V2.0, provided by the NOAA/OAR/ESRL Physical Sciences Division, Boulder, Colorado, USA and available at http://www.esrl.noaa.gov/psd/ [29]. The used method is the dual-constraint spatiotemporal clustering approach [30] to find abnormal marine variations. The Multivariate ENSO Index (MEI) is used to conduct a correlation analysis with the process objects [31].

Finally, 219 process objects, including 653 sequence objects and 2634 state objects, are obtained, and a process-oriented marine spatiotemporal graph database, based on Neo4j, named *PomSTGDB*, is built to store them. Among these process objects, several of them have similar dynamic characteristics and have strong relationships with ENSO [30,32]. From the *PomSTGDB*, we know not only the marine phenomena as well as its changes but also the cause of the changes in the developing, merging, splitting, or merging-splitting behaviors. In addition, the occurrence of this behavior will strengthen or weaken an ENSO event. Figure 14 illustrates the evolution of an abnormal sea surface temperature variation from Feb 1997 to Jan 1999. The dynamic process includes 8 evolution sequences and 38 instantaneous states.

In Figure 14, in February 1997, Sequence 1 is generated in the Eastern Pacific Ocean and expands in space from March to May, and Sequence 2 is generated in the Central Pacific Ocean and expands from March to June. In May, Sequence 3 and Sequence 4 are generated in the Eastern Pacific Ocean. In June, Sequence 1 and Sequence 4 merge into Sequence 5. In July, Sequence 1, Sequence 3, and Sequence 5 merge into Sequence 6 in the Central Eastern Pacific Ocean, where Sequence 6 continues to develop from July 1997 to February 1998. In March 1998, Sequence 6 splits Sequence 7 in the Central Eastern Pacific Ocean, and Sequence 8 splits in the Eastern Pacific Ocean. From April, Sequence 7 from the Central Pacific Ocean begins to shrink in space and disappears in January 1999 in the Eastern Pacific Ocean, and Sequence 8 decreases and disappears in August 1998 in the Eastern Pacific Ocean.

The generation, development, merging, splitting, and disappearance of the abnormal marine process are generally consistent with the evolution of an El Niño event, occurring between 1998 and 1999, as shown as Figure 15. When Sequence 1 and Sequence 2 are being generated, the El Niño event is also about to begin, and at the time of Sequence 1, Sequence 3, and Sequence 5 merging in a space, the strength of the El Niño event reaches its maximum. During the period of the development of Sequence 6, the El Niño event remains stable, and when the process splits and disappears after three months, the El Niño event begins to decrease and then disappears over time.

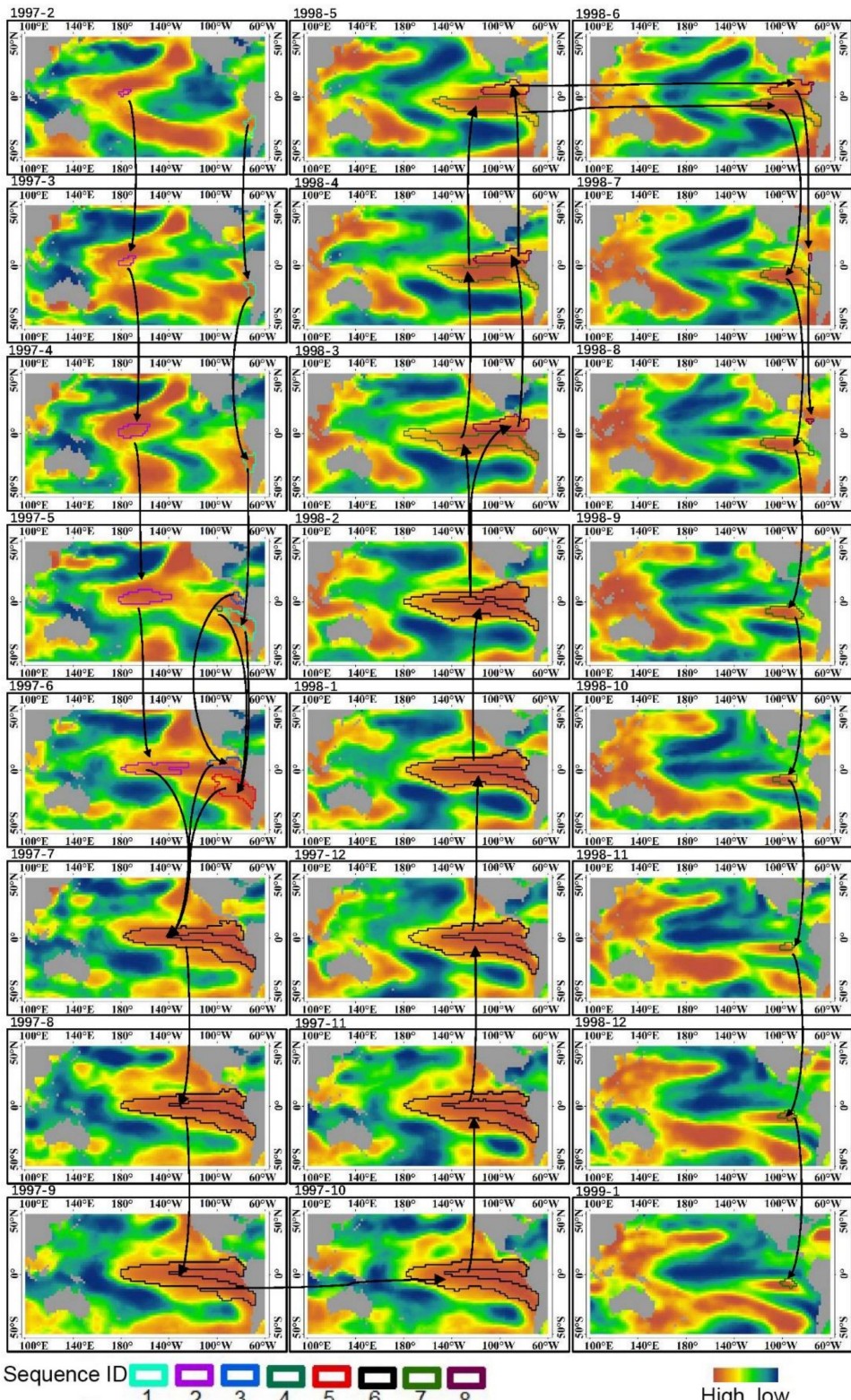

**Figure 14.** The evolution of an abnormal sea surface temperature variation and its evolutions: The background is made up of monthly anomalies of the sea surface temperature; different colors of the Sequence ID represent different evolution sequences; and the black arrow represents an evolution, developing, merging, or splitting relationship.

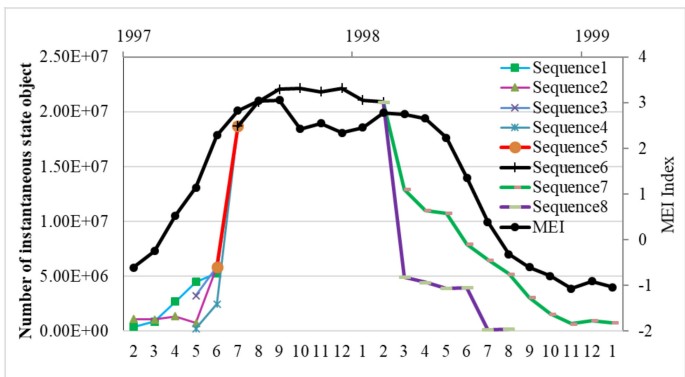

**Figure 15.** The relationship between the evolution of the mined process and the El Niño event.

## 5. Conclusions

The investigation of the evolution of dynamic geographic phenomena requires an efficient approach since spatiotemporal data need to be linked to a reasoning mechanism. To address this issue, this paper proposed a process-oriented two-tier graph model to represent and store dynamic geographic phenomena. A simulated dataset and a real dataset from remote sensing products were used to test this model. The main conclusions and contributions of this study are as follows. (1) The *PoTGM* consists of four types of nodes, i.e., a process node, a sequence node, a linked node, and a state node, and two types of edges, i.e., an including edge and an evolution edge. This not only ensures that the dynamic geographic phenomena are represented and stored but also provides a mechanism for addressing the spatial evolution.

(2) The including relationships achieve proximity in space and homogeneity in thematic attributes; thus, the *PoTGM* could transform the dynamic phenomenon from a geographic process to instantaneous states, and vice versa. As the evolution relationships achieve continuity in time, the *PoTGM* could recall the history, depict the current, and predict the future of dynamic phenomena.

(3) As for the database storage capacity and object querying, the Oracle spatial database performs better than *PomSTGDB*; while regarding the relationship querying, *PomSTGDB* performs one-order magnitude better than Oracle. In relation to the dynamic geographic phenomena, the evolution relationships obtain much more attention than the phenomena themselves.

(4) Using *PomSTGDB*, we address the process, evolution sequences, and instantaneous states of an abnormal sea surface temperature evolution and the spatial evolution as well, i.e., developing, merging, and splitting. In addition, these behaviors will strengthen or weaken the intensity of ENSO events. This not only speaks to the effectiveness of *PoTGM* but may itself serve as an important reference for research on the mutual response and driving mechanisms behind global climate change and abnormal marine variations.

In a previous work, we designed a dual-representing framework with a "process-phase-sequence -state" [16]. However, in a physical world, it is difficult to identify the sequence and the phase. In addition, Xue et al. focused on the process objects and the including relationships among them and ignored the evolution relationships [16]. This paper contributes to the field of geographic information science research as follows. 1) The *PoTGM* provides a routine for analyzing the dynamic behaviors of geographic phenomena by storing the objects as well as their evolution relationships; 2) an evolution of sea surface temperature anomalies in the Pacific Ocean is addressed, and the results could provide new references for global change research.

The semantic of the "geographic process-evolution sequence-instantaneous state" makes it possible to represent and store the phenomena in an evolution scale of dynamic characteristics, not a time scale of data acquisition. The evolution scale takes the dynamic phenomenon during its lifespan as a storage unit, which ensures proximity in space, continuity in time, and homogeneity in

thematic attributes. Compared to the separation of time and space, process-oriented storage could improve the investigative capacities relating to dynamic phenomena, which need further study.

**Author Contributions:** Conceptualization, C.X. and F.S.; Data curation, J.L.; Formal analysis, C.W.; Funding acquisition, C.X.; Methodology, C.X.; project administration, C.X.; software, C.W.; Validation, J.L.; Writing—original draft, C.X.; Writing—review and editing, F.S.

**Funding:** This research was supported by the National key research and development program of China (No. 2016YFA0600304, 2017YFB0503605), by the Strategic Priority Research Program of the Chinese Academy of Sciences (No. XDA19060103), and by the National Natural Science Foundation of China (No. 41671401).

**Acknowledgments:** We thank the data working group of the National Oceanic and Atmospheric Administration/The Office of Oceanic and Atmospheric Research/Earth System Research Laboratory/Physical Sciences Division for providing the research data.

**Conflicts of Interest:** The authors declare no conflict of interest.

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
