# Peer review of "A Novel Process-Oriented Graph Storage for Dynamic Geographic Phenomena"

_ijgi, doi:10.3390/ijgi8020100_

Round 1

Reviewer 1 Report

General comments:
Overall this is a good research work.
I would advise the authors to take a look at the following online resource (English for Writing Research Papers Useful Phrases ) from springer, to help out in the revision of this manuscript and future manuscripts: https://t.co/kX6LTRrGbz
The authors need to expand a bit more the legend of each Figure, keeping in mind that the legends should be self-explaining and provide sufficient information to the reader without referring to the text in the manuscript.
The authors should also consider to ask a colleague not directly involved in the writing of the manuscript to read it after the suggested changes have been made.

Comments & Suggestions:

Line 31
Replace dynamics with dynamic

Line 31&32
Re-write the sentence highlighted in blue

Line 37&38
Re-write the start of the sentence (highlighted in blue)

Line 42&43
I don’t know what the authors mean with composited model, I didn’t found any reference concerning spatiotemporal composited model.

Line 49
Replace “which has incapability” with “not capable”

Line 51 & rest of text
Replace “ evolution behaviors” with “spatial evolution” or something the authors might find more appropriate

Line 56
Try to replace “hot issue” with something else, from my point of view this type of wording should be avoided in scientific writing.

Line 59
Explain in one simple sentence what is the “logistical design”

Line 60
The citation/reference seems that has not been taken by the software used.

Line 64&65
Re-write the sentence highlighted in blue

Line 66
Re-write the sentence highlighted in blue

Line 79&80
Re-write the sentence highlighted in blue

Line 81 to 85
Split the sentence in 2 or 3 smaller sentence. The sentence is 2 big and the reader gets easily lost.

Line 87
At the end of the sentence had a small sentence describing what you (the authors) considered has uniform semantic, with the redefinition or by giving an example.

Line 88
Re-write the text highlighted in blue. Something like “A common way to store such types of dynamic graphical entities”.

Line 90&91
The text highlighted in yellow can be deleted.

Line 93
Re-write the text highlighted in blue.

Line 108to 111
Try to expand a bit more, and add a table with the numerated issues.

Line 111
Delete “a”

In 2.1.Dynamic geographic Phenomena it would be interesting to have. A ”real world” example made by the author in figure format. The can also be a citation (Figure) from another author.

In 2.3. Relationships of the Geographic Process
I would advise the authors to expand a bit more on the four types of evolution relationships given from Line 166 to 177. It’s the focus of this paper, so this should be well explained before moving to 3. Process-Oriented Two-Tier Graph Model.

3. Process-Oriented Two-Tier Graph Model.
Line 187&188 give examples of the three kinds of relationships.

Line 196
Remove “as”

Line 213&214
Re-write the text highlighted in blue

Line 230&231
Re-write the text highlighted in blue

Since From line 234 to line 238 you are describing the action’s made by each Node, I suggest the following changes:
Line 234
Remove “are to”

Line 235
Replace “is to find” with “finds”

Line 236
Replace “And EvolutionEdge is to find“ with “The EvolutionEdge finds”

Line 240
Remove “ common”

Line 244
After the highlighted text add 1 or 2 examples “, such as …”.

Line 244
Change the order of the text to something like:
“Fig.7 shows a schema based on Neo4j of PoTGM, using the doubly linked list.”.

Line 249
Remove “s”

In 4.1. Simulated Data and Performance Analysis
Move Figure 8 to after the first paragraph, before Figure 9.

Line 262
Re-write the text highlighted in blue

Line 264
Remove text highlife in yellow

Line 264 to 266
Re-write the sentence.

Line 273
Remove text highlighted in yellow
Replace “does” with “performs”

Line 274
Replace “a” with “the”

Line 276
I think the authors meant to say “above 80%” instead of what is written (80% above).

Line 277
Re-write the text highlighted in blue

Line 283
Replace “a” with “the”

Line 284
Re-write the text highlighted in blue

From lines 292 to 306 is the first time I see “relationship depth”, but I don’t know what the authors define or consider as “relationship depth”. Please expand or use another name instead of depth.

Move figure 13 to after the text describing the content of the figure. Expand Figure 13 legend with for example what the black arrows, black polygons and green polygons represent.

 Line 318
Replace the text highlighted in blue by “, and”

Line 350
Re-write highlighted text in blue to something like
“The investigation of the evolution of dynamic geographic phenomenon’s requires an efficient approach, since spatiotemporal data needs to be linked to a reasoning mechanism.”

Line 376
Replace “GIScience “ With Geographic Information System.

Line 375
Check the citation/reference, probably it ha snot been properly considered by the software used.

Line 378&380
Re-write the text highlighted in blue.

Author Response

Comments 1: I would advise the authors to take a look at the following online resource (English for Writing Research Papers Useful Phrases ) from springer, to help out in the revision of this manuscript and future manuscripts: https://t.co/kX6LTRrGbz.

Answer: Thanks for your meaningful suggestions for both this manuscript and our further work. For improving the quality and readability of this manuscript, we invited a native-English-speaking scientific editor supporting by MDPI English editing to rewrite the revised manuscript.

Comments 2: The authors need to expand a bit more the legend of each Figure, keeping in mind that the legends should be self-explaining and provide sufficient information to the reader without referring to the text in the manuscript.

Answer: The reviewer is right. Thanks for your suggestions. The revised manuscript expands the description of Figure 4-6, Figure 9 and Figure 13 .

Comments 3: The authors should also consider to ask a colleague not directly involved in the writing of the manuscript to read it after the suggested changes have been made.

Answer: Thanks. For improving the quality and readability of this manuscript, we invited a native-English-speaking scientific editor supporting by MDPI English editing to rewrite the revised manuscript.

Comments 4: The other comments about the spells, grammars and expressions listed in the attached file.

Answer: We would like to express our faithful appreciation and thanks for your hard work. Exception of two references, we addressed all the comments, and followed them to revised our manuscript.

The reviewer is right. To the known of our knowledge, the reference of Yuan (2001) has not been taken by the commercial software used. Another reference is our previous work (Xue et al., 2012), although the process model also has not been taken by the commercial software used, we designed a prototype software to test our model by using Oracle spatial database.

1.        Yuan. M. Representing complex geographic phenomena in GIS. Cartography and Geographic Information Science, 2001, 28,2,83-96.

2.        Xue, C.J.; Dong, Q.; Xie, J. Marine spatio-temporal process semantics and its application-taking the El Nino Southern Oscilation process and Chinese rainfall anomaly as an example. Acta Oceanol. Sin. 2012,31,2,16-24.

Reviewer 2 Report

There are many spelling errors, mix-ups of singular and plural and strange phrasings in the text. I would recommend that you at least use a spell checker. Additional proofreading or English editing service would also be recommendable.

I felt a bit lost in section 2.1. You use terms that are not explained until section 2.2. Your definitions of production, development and death sequence should be placed in section 2.1.

Concerning figure 3: The used type of legend can be very confusing. It looks like the arrows in the legend represent states in the timeline (because they are directly underneath the timeline). Therefore, the legend should be placed either on the right side or on the left side of the image.

At the end of the conclusions section, you write that  "the process-oriented storage will improve the capabilities of exploring the dynamic phenomena..". Such a claim would require user studies, which are not mentioned here. Therefore, you should replace "will" by "could".

Author Response

Comments 1: There are many spelling errors, mix-ups of singular and plural and strange phrasings in the text. I would recommend that you at least use a spell checker. Additional proofreading or English editing service would also be recommendable

Answer: Thanks a lot for your suggestions. For improving the quality and readability of this manuscript, we invited a native-English-speaking scientific editor supporting by MDPI English editing to rewrite the revised manuscript.

.

Comments 2: I felt a bit lost in section 2.1. You use terms that are not explained until section 2.2. Your definitions of production, development and death sequence should be placed in section 2.1.

Answer: Thanks for your comments. Section 2.1 only analyzes the properties of dynamic geographical phenomena, and discusses the differences with the discrete change, which doesnot begin to give the definition of process. From Section 2.2, this paper begins to give the process semantic, including the definitions and relationships. After serious consideration with other authors, we think the structure of Section 2.1 and 2.2 might be more right. If the reviewer still believes that the definitions of production, development and death sequence should be placed in section 2.1, we will rewrite Section 2.1 and 2.2.

Comments 3: Concerning figure 3: The used type of legend can be very confusing. It looks like the arrows in the legend represent states in the timeline (because they are directly underneath the timeline). Therefore, the legend should be placed either on the right side or on the left side of the image.

Answer: So sorry to make the reviewer unclear about Figure 3. Thanks for your suggestion, and follow it we redraw Figure 3.

Comments 4: At the end of the conclusions section, you write that  "the process-oriented storage will improve the capabilities of exploring the dynamic phenomena..". Such a claim would require user studies, which are not mentioned here. Therefore, you should replace "will" by "could".

Answer: Thanks. The sentence is rewritten by replacing “will” by “could”.

Reviewer 3 Report

In my opinion, this work makes a very interesting paper to IGJI journal as it contributes to the search for solutions to the problems currently posed when dealing with the management of spatio-temporal data. The research design is appropriate, the methodology followed clearly explained and the conclusions can be useful for future work in this area. Therefore, I recommend its publication, although, despite not being an English native myself, I believe that a revision of the use of English is necessary in both grammar and spelling.

The structure of the paper is adequate and it presents the arguments clearly. It begins explaining the dynamic nature of geographic processes and raises the need for models adapted to the temporal nature of spatial data, to allow their analysis, management and representation. The authors make a good review of the state of the art from the eighties to the present, with the solutions proposed to the problem and the shortcomes found in each case. The references are adequate and the bibliography seems complete to me.

To track the evolution of spatial objects, an efficient approach is needed to link spatio-temporal data with a reasoning mechanism. To address this problem, the authors propose a two-level process-oriented graph model (named as PoTGM) to represent and store dynamic geographic phenomena. The first level graph stores the processes and the second level the sequence and states. The authors used Neo4j, a graph database to implement their model, which was tested with two case studies, both simulated and real. The results are discussed and several conclusions drawn.

Author Response

Comments 1: despite not being an English native myself, I believe that a revision of the use of English is necessary in both grammar and spelling.

Answer: Thanks for your suggestions. For improving the quality and readability of this manuscript, we invited a native-English-speaking scientific editor supporting by MDPI English editing to rewrite the revised manuscript.

Reviewer 4 Report

The work proposed in the paper sounds really interesting and useful. However, some comments are mandatory.

First, the introduction section lacks of a figure illustrating the process-oriented data model. This would be especially useful for those who are not used to this kind of model.

Then, the authors must review the last paragraph of the introduction. First, it is not true to argue that Oracle Spatial, SQL Server, PostgreSQL are “object-oriented spatial databases”. These are “object-relational databases”. Furthermore, the statement that these spatial databases do not focus on the spatial relationships is quite direct, and should be reconsidered. The spatial relationships are actually considered in such databases based on foreign keys, and logistical relationship tables. Eventually, the usefulness of graph-based databases (compared to object-relational databases) should be more developed.

The section 3 (Process-Oriented Two-Tier Graph Model) should incorporate a concrete example illustrating the theoretical concepts. This remains too abstract. Maybe, a small example with the El-Nino-Southern Oscillation. Finally, in page 7, the authors should explain the index-free adjacency.

Eventually, some misprints in the paper. Page 5: splitting-merging relationship. The relationship between them is a splitting-merging relationship, not development relationship. Page 8: An edge stores two pointers, one points to the previous node, denoted as NextEdgeID, the other points to the next node, denoted as PreEdgeID. I think that the authors mean the reverse for NextEdgeID and PreEdgeID.

Author Response

Nan

Round 2

Reviewer 4 Report

First, the introduction section still lacks of a figure illustrating the process-oriented data model. 

Then, it is still not true to argue that Oracle Spatial, SQL Server, PostgreSQL are “object-oriented spatial databases”. These are “object-relational databases”. Furthermore, the statement that these spatial databases do not focus on the spatial relationships is quite direct, and should be reconsidered. The spatial relationships are actually considered in such databases based on foreign keys, and logistical relationship tables. Eventually, the usefulness of graph-based (compared to object-relational databases) databases should be more developed.

The section 3 (Process-Oriented Two-Tier Graph Model) remains too abstract. Maybe, a small example with the El-Nino-Southern Oscillation. Finally, in page 7, the authors should explain the index-free adjacency.

Eventually, some misprints are still in the paper. Page 5: splitting-merging relationship. The relationship between them is a splitting-merging relationship, not development relationship. Page 8: An edge stores two pointers, one points to the previous node, denoted as NextEdgeID, the other points to the next node, denoted as PreEdgeID. I think that the authors mean the reverse for NextEdgeID and PreEdgeID.

Author Response

Firstly, we would give our apologies to you for not replying your comments last time. When receiving your comments, we have no enough time to revise our manuscript.  Now, we have carefully studied your comments and have made corrections to the manuscript that we hope will satisfy your requirements. The details are as follows.

Comments 1: First, the introduction section lacks of a figure illustrating the process-oriented data model. This would be especially useful for those who are not used to this kind of model.

Answer: Thanks for your comments. Your suggestion will improve the readabilities of our manuscript, especially for those who are not used to this kind of model. Line 58 to Line 75 addresses the research status of the process-oriented data model, showing that there are not a uniform data model. E.g. Yuan (2001) designed a hierarchical framework with a zone-sequence-process-event; Xue et al. (2012) proposed a process-oriented dual representing framework, with a semantics of hierarchical abstraction and an object of being included by level, i.e., a process-sequence-phase-state. Thus, we have challenges of using a figure to illustrate the process-oriented data model in Introduction Section.

For improving the readabilities of our manuscript, we draw a figure to describe the process-oriented data model in Section 2.2.

Comments 2: Then, the authors must review the last paragraph of the introduction. First, it is not true to argue that Oracle Spatial, SQL Server, PostgreSQL are “object-oriented spatial databases”. These are “object-relational databases”. Furthermore, the statement that these spatial databases do not focus on the spatial relationships is quite direct, and should be reconsidered. The spatial relationships are actually considered in such databases based on foreign keys, and logistical relationship tables.

Answer: You are right, thanks. These descriptions are not serious. We want to represent that the object-relational databases focus on the storage and management of objects, not their spatial changes. Thanks for your comments. These sentences are rewritten.

Comments 3: Eventually, the usefulness of graph-based databases (compared to object-relational databases) should be more developed.

Answer: Thank you for your suggestions. The revised manuscript expands the descriptions of the usefulness of graph-based databases in the second to the last paragraph.

Comment 4: The section 3 (Process-Oriented Two-Tier Graph Model) should incorporate a concrete example illustrating the theoretical concepts. This remains too abstract. Maybe, a small example with the El-Nino-Southern Oscillation.

Answer: Thank you for your suggestions. The revised manuscript uses the simulated geographic process as an example, and adds one paragraph to illustrate the theoretical concept of the process-oriented two-tier graph model in the end of Section 3.2.

Comment 5: Finally, in page 7, the authors should explain the index-free adjacency.

Answer: Thank you for your suggestions. We give a full explains about the index-free adjacency in the first paragraph in Section 3.2.

Comment 6: Eventually, some misprints in the paper. Page 5: splitting-merging relationship. The relationship between them is a splitting-merging relationship, not development relationship. Page 8: An edge stores two pointers, one points to the previous node, denoted as NextEdgeID, the other points to the next node, denoted as PreEdgeID. I think that the authors mean the reverse for NextEdgeID and PreEdgeID.

Answer: Thank you for your suggestions. For the first error, we revise it from “a development relationship to a splitting-merging relationship”. It is our carelessness. For the second comments, an edge pointing to the previous node is behind the node, so we denote its ID as NextEdgeID. And the edge pointing to the next node is inverse. Thus, the NextEdgeID and PreEdgeID are suitable.
